# Placental Ultrasonographical Findings during SARS-CoV-2 Infection

**DOI:** 10.3390/diagnostics12040974

**Published:** 2022-04-13

**Authors:** Sotirios Sotiriou, Athina A. Samara, Iokasti-Apostolia Tsiamalou, Christos Donoudis, Eleni Seviloglou, Chara Skentou, Antonios Garas, Alexandros Daponte

**Affiliations:** 1Department of Embryology, Faculty of Medicine, University of Thessaly, 41110 Larissa, Greece; sotiriousoti@yahoo.gr (S.S.); skentou@gmail.com (C.S.); 2Department of Obstetrics and Gynecology, University Hospital of Larissa, 41110 Larissa, Greece; iokasti94@hotmail.com (I.-A.T.); xdonoudis@gmail.com (C.D.); eseviloglou@gmail.com (E.S.); garasant@yahoo.gr (A.G.); daponte@uth.gr (A.D.)

**Keywords:** COVID-19, placenta, ultrasound, malperfusion, pregnancy

## Abstract

Infection with SARS-CoV-2 virus (COVID-19) during pregnancy has been associated with several complications. Increasing evidence suggests that COVID-19 infection leaves tell-tale signs of placental injury. During ultrasound examination and placental evaluation of COVID-19 infected pregnancies, we recorded signs of placental involvement, with findings indicating malperfusion, chorangiosis, deciduitis, and subchorionitis. Early detection of placental damage through the use of specific ultrasound findings could indicate which pregnancies are at increased risk for complications.

## 1. Introduction

Pregnancy has always been associated with increased susceptibility to respiratory infections and subsequently higher morbidity and mortality [1]. In a similar manner, infection with SARS-CoV-2 virus (COVID-19) during pregnancy has been associated with several complications, including increased rates of spontaneous and iatrogenic preterm labor, with rates of preterm birth at 17% compared to 8–10% reported in the general population [2,3]. COVID-19 is considered a systematic disease that affects many organs from different systems, including the placenta [1]. Since the start of the COVID-19 pandemic in 2019, several studies have attempted to assess the transplacental virus transmission and its effects on pregnancy outcomes [3,4,5].

COVID-19 infection results in both pulmonary and extra-pulmonary symptomatology [6]. While multi-organ involvement of this disease entity has been reported extensively in the literature, data related to the obstetric population are still unclear [7]. Common symptoms of COVID-19 include common cold symptoms, a cough, the loss of smell and taste, headache, and fever [8]. A recently published study reports the rate of asymptomatic infections to be 35%, while the rate of severe or critical cases is 4.6% [8]. During the Omicron wave, fully vaccinated pregnant women with COVID-19 experienced milder disease and were less likely to require oxygen supplementation or intensive care compared with their unvaccinated counterparts [9].

Diagnostic ultrasound has been used in clinical obstetrics for the last half a century. However, placental examination appears to be treated with less attention than fetal examination [10]. Examination of the placenta plays a critical role in the assessment of normal and abnormal pregnancies; this evaluation includes the morphology, anatomy, location, implantation, anomaly, size, and pulsed Doppler ultrasound assessment of the placenta [11].

Increasing evidence suggests that COVID-19 infection leaves tell-tale signs of placental injuries [1]. In trophoblasts of infected pregnant patients, placental features such as fetal vascular malperfusion, arteriopathy and inflammation, perivillous fibrin deposition, and fetal vessel thrombosis have been reported [12]. In this context, early detection of placental damage with the use of specific ultrasound findings may indicate which pregnancies are at increased risk for complications, such as pre-eclampsia, intrauterine growth retardation (IUGR), and even sudden fetal death [13].

Herein, we report our experience of a tertiary referral for COVID-19 infected pregnancies from a tertiary hospital’s Department of Prenatal Ultrasound.

## 2. Methods

The present cohort study included pregnant women consecutively testing positive via a SARS-CoV-2 real-time polymerase chain reaction (RT-PCR) molecular test during a three-month period (September 2021–December 2021). Patients were divided in two groups: patients requiring hospital admission based on their symptoms (group 1), and patients experiencing minor symptoms not requiring hospitalization (group 2). All participants underwent an ultrasound examination and placental sonographic evaluation in the Prenatal Ultrasound Department.

Clinical symptoms for admission were in accordance with the local COVID-19 protocols, including the need for respiratory support, a fever above 38 °C for more than 24 h, confirmed pneumonia, etc. To avoid systematic bias, in this study, we excluded pregnant women infected by COVID-19 presenting asymptomatically who were hospitalized for obstetrical indications.

Informed consent was obtained from all participants. Experimental therapeutic protocols are not applicable in this study. All data were analyzed anonymously using code numbers with respect to the patient’s privacy and collected in the context of routine diagnostic and therapeutic procedures.

Our department created a scoring system for all placental ultrasonographic findings that were observed, to be simple and reproducible for further research (Table 1). The scoring system has been used retrospectively by the same team. Τhe same standard ultrasound settings were used in all cases. Furthermore, all cases (hospitalized or not) were followed-up in our unit as part of the standard obstetrical care. Each finding observed is associated with 1 point. The total score, ranging from 0 to 3, is calculated by the sum of the following points.

## 3. Results

In total, 40 pregnant women were included in the present study; 10 required hospital admission (group 1) and 30 remained at home (group 2). There was no statistically significant difference regarding pregnancy characteristics between the groups.

All women in group 1 had at least 2 points in the scoring system (at least two ultrasonographic findings), while women in group 2 had 0–1 points in the scoring system (Table 2).

One woman was admitted to the intensive-care unit (ICU) for intubation during the 26th week of pregnancy based on her low oxygen levels. An additional 9 women were admitted to the ICU between 13 to 34 weeks of gestation, requiring hospital support and follow up for 5–7 days. All women recovered completely and had a negative test via RT-PCR. A total of 30 women with only mild symptoms from 18 to 31 gestational weeks remained at home with instructions for care.

Ultrasonographic findings:

During the ultrasound examination and placental evaluation, we recorded signs of placental involvement with findings indicating malperfusion, chorangiosis, deciduitis, and subchorionitis, such as fibrin deposits, lakes, subchorionic fibrin deposition, and thrombosis (Figure 1 and Figure 2).

Placental findings were present in more than 80% of group 1 cases. The most prominent findings were subchorionic fibrin deposition and placental lakes with perivillous fibrin deposits of at least 30% of the placental mass, compared to only 25% of group 2 cases.

The present cohort study includes data recently collected between September and December 2021, and most of the pregnancies included have not been terminated yet. Furthermore, the present data are preliminary regarding neonatal birthweight; a possible association between clinical symptomatology of COVID-19 infection and adverse pregnancy outcomes will be investigated.

## 4. Discussion

Similar to other RNA viral infections such as Zika virus, cytomegalovirus, and dengue virus during pregnancy, SARS-CoV-2 placental infection is associated with chronic inflammatory pathologies indicating lymphohistiocytic villitis, chronic histiocytic intervillositis, and chronic deciduitis [1,14]. In our series, more than 80% of placentas from infected pregnancies with severe clinical symptoms had sonographic findings indicating histopathological abnormalities. Excessive fibrin, calcifications, and subchorionic fibrin deposition assuming chorangiosis were significantly higher than in SARS-CoV-2-positive women with only minor symptoms. These abnormalities can lead to trophoblast necrosis, massive fibrin deposition, and decreased exchange between mother and fetus in a short timeframe [2].

Since SARS-CoV-2 has been detected in the placenta of pregnant women infected by COVID-19, great interest has been seen regarding placenta physiology. Angiotensin-converting enzyme 2 (ACE2) receptor is expressed both by the virus and by numerous endothelia as well [15]. The virus enters cells by binding the spike protein to the receptor of the ACE2, a receptor present in the lung and small intestine epithelia, as in arterial and venous endothelial cells in all organs [16,17]. In the placenta, ACE2 is expressed in the stromal and perivascular cells of decidua, fetal placental vessels, and in cytotrophoblast [15].

A systemic thrombotic and microvascular placental injury syndrome caused by SARS-CoV-2 infection includes histology findings with focal avascular villi and thrombi in larger fetal vessels [18,19]. The presence of placental injury findings may be a result of systemic effects of the virus rather than the virus itself [19]. Fetal vascular thrombi are developed due to various combinations of three risk factors (Virchow’s triad): stasis, hypercoagulability, and endothelial or vessel wall damage [19]. Degeneration and loss of capillaries can occur secondary to thrombosis in the chorionic plate. Chorioangiosis occurs as a result of the initial ischaemic phase and consequently leads to karyorrhexis of fetal endothelium with resultant capillary wall disruption and spillage of necrotic cell fragments into the villous stroma with resultant degeneration. Vascular ectasia can occur secondary to increased intraluminal pressure and degeneration [19,20]. The significant association of chorioangiosis, intramural fibrin deposition, and vascular ectasia of fetal vessels with SARS-CoV-2 positive pregnancies indicates fetal vascular malperfusion in the placenta of the infected pregnancies [21].

Viraemia may trigger a “cytokine storm” following overwhelmed maternal immune response to SARS-CoV-2 infection, with a surge in pro-inflammatory cytokines including interferon-G (IFN-G), interleukin-2 (IL-2), IL-6, IL-7, IL-10, and tumor necrosis factor (TNF). Other studies posit that the increased production of inflammatory biomarkers may be triggered by placental hypoperfusion/ischemia due to maternal hypoxia, following severe COVID-19 infection [22,23]. The uncontrolled release of inflammatory cytokines will further exaggerate the maternal immune system and link to the occurrence of placental damage, fetal growth restriction, abortion, or preterm labor [24].

Fetal vascular malperfusion (FVM) is a feature manifested in placentas with diminished vascular supply, reported in studies with COVID-19-affected placentas [25,26]. Fetal vascular thrombosis, abnormal cord insertion, hypercoiling of the umbilical cord, and a maternal hypercoagulable state are among conditions associated with FVM [27]. The endotheliotropic behavior of SARS-CoV-2 via the ACE2 receptor on endothelial cells makes it prone to causing vascular endothelial dysfunction, leading to a complement-induced coagulopathy state in COVID-19-infected patients, and therefore susceptibility to microthrombi formation [28].

To the best of our knowledge, the present study is one of the pioneer studies that reports specific placental ultrasonographic findings attributed to COVID-19 infection. However, prior to the appraisal of our results, several limitations must be considered. A major limitation is the relatively small number of patients recruited from a single department. Furthermore, the retrospective use of the scoring system remains a major limitation of this study. Larger cohort studies investigating a possible association between ultrasound placental findings and pregnancy outcomes are needed.

In conclusion, early detection of placental damage through the use of specific ultrasound findings could indicate pregnancies that are at an increased risk of complications. Ultrasound findings including placental lakes, fibrin deposits, and subchorionic edema can be used as a diagnostic tool for possible fetal vascular malperfusion, as well as early signs of adverse pregnancy outcomes requiring closer antenatal follow-up. As the pandemic progresses, further research based on large populations is required to come to a safer conclusion.

## Figures and Tables

**Figure 1 diagnostics-12-00974-f001:**
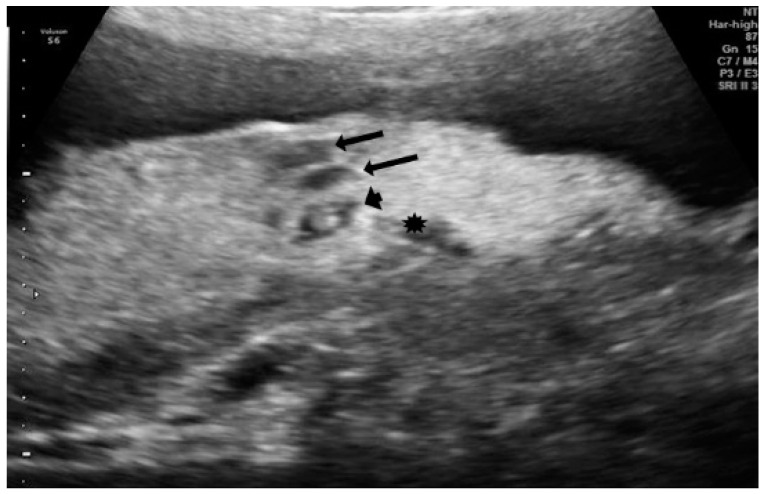
Ultrasonographic placental image: placental lakes (arrow), fibrin deposits, signs of thrombosis (arrowhead), and subplacental blood pools are observed.

**Figure 2 diagnostics-12-00974-f002:**
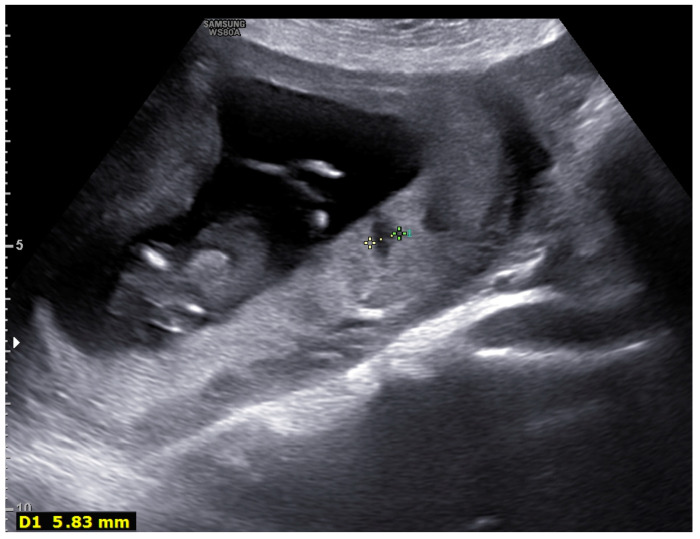
Ultrasonographic placental image of a twelve-week pregnancy: blood pools (max diameter 5.8 mm) are observed.

**Table 1 diagnostics-12-00974-t001:** Scoring system for placental ultrasonographic findings: each finding is associated with 1 point. The total score, ranging from 0 to 3, is calculated by the sum of the following points.

Scoring System for Placental Ultrasonographic Findings	
0	Normal findings
1	Lakes > 30%
1	Subchorionic edema
1	Fibrin deposits (thrombosis, calcifications)

**Table 2 diagnostics-12-00974-t002:** Number of cases with placental ultrasonographical findings in each group.

U/S Findings/Group	Hospital Admission N (%)	Remaining at Home N (%)
Normal findings	0 (0%)	18 (60%)
Lakes > 30%	7 (70%)	6 (20%)
Subchorionic edema	6 (60%)	2 (7%)
Fibrin deposits (thrombosis, calcifications)	6 (60%)	4 (13%)

## Data Availability

Data are availible upon reasonable request.

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
