# Peer review of "Placental Ultrasonographical Findings during SARS-CoV-2 Infection"

_diagnostics, 2022, doi:10.3390/diagnostics12040974_

Round 1
Reviewer 1 Report
Dear authors,
The topic is interesting and actual, however you need to explore some itens:
introdution: this topic must be more exaustive, about the obstetric ultrasound and placent evaluation and also about COVID-19 in obstetric conditions.
Methods: it is important to refer the ethic comission and informed consent. Is The Scoring system for placental ultrasonographic findings adapted from some author? if not why do you use this scoring?
Discussion: must be more extensive, with results comparasion. The limitations must be in this topic.
Conclusion: must be destacated.
Author Response
Reviewer comment: The topic is interesting and actual, however you need to explore some itens:
introdution: this topic must be more exaustive, about the obstetric ultrasound and placent evaluation and also about COVID-19 in obstetric conditions.
Answer: We would like to thank the reviewer for the well-stated comments. In line with these comments, two paragraphs have been added analyzing the above-mentioned issues in introduction section.
Reviewer comment: Methods: it is important to refer the ethic comission and informed consent.
Answer: We would like to thank the reviewer for the well-stated comments. Informed consent was obtained from all participants. According to local Ethic Committee the approval was not obligatory for our observational study because there were no interventional actions. A new paragraph with the above mentioned statements was added.
Reviewer comment: Is The Scoring system for placental ultrasonographic findings adapted from some author? if not why do you use this scoring?
Answer :The scoring system was not adopted from other authors. The system was created by our team in order to be simple and reproducible for further research.
Reviewer comment: Discussion: must be more extensive, with results comparasion. The limitations must be in this topic.
Answer: We would like to thank the reviewer for the well-stated comments. Three new paragraphs were added in the discussion section.
Reviewer comment Conclusion: must be destacated.
Answer: A conclusion statement was added.
Reviewer 2 Report
I have read this paper with great interest. Although topical report on pregnancy related COVID outcome data and biomarkers are for sure useful, but the current version has in my opinion major limitations. We should have more information on the completeness of the cohort, and on the character of the study design, and we have no data on outcome so that the suggestion to screen for abnormalities is disconnected from outcome data.
Specific
Line 24-25: are you sure ‘not considered’ a systematic disease that affects many organs from different systems… : do you mean ‘considered’ ?
Line 32-33: I think that there are also reports on sudden fetal death associated with covid ?
Methods
Can you comment on the indication for admission ? where these solely COVID driven, or where there also pregnancy related indications ? where all cases included ?
Was the scoring system used prospectively, or retrospectively, and the assessors were likely not blinded for covid, hospitalization and outcome ? and how were settings handled with several ultrasounds (the worst imaging, the highest score, progressing pregnancy). How should we compare the findings in hospitalized and non-hospitalized cases (80 and 25 %), to ‘reference datasets over gestational age’ ?
Are there any data on pregnancy outcome, and on APO of the placenta after delivery ?
Editing:
I think that the reference numbers should be adapted to be in line with guidelines of the journal.
Author Response
Reviewer comment: I have read this paper with great interest. Although topical report on pregnancy related COVID outcome data and biomarkers are for sure useful, but the current version has in my opinion major limitations. We should have more information on the completeness of the cohort, and on the character of the study design, and we have no data on outcome so that the suggestion to screen for abnormalities is disconnected from outcome data.
Answer: We would like to thank the reviewer for the well-stated comments. The present study is an observational cohort study including consecutive patients. The present cohort study includes recently collected date between September to December 2021 and the most of the pregnancies included have not been terminated yet. Furthermore, the present data are preliminary regarding neonatal birthweight and a possible association with IUGR. Based on those preliminary data, there is no to association regarding the above parametres, but we are under process of collection more data.
Reviewer comment: Specific
Line 24-25: are you sure ‘not considered’ a systematic disease that affects many organs from different systems… : do you mean ‘considered’ ?
Answer: It was a typing error that have been corrected. Thank you for the comment
Reviewer comment: Line 32-33: I think that there are also reports on sudden fetal death associated with covid ?
Answer: Thank you for the comment. The above-mentioned complication was added.
Reviewer comment: Methods
Can you comment on the indication for admission ? where these solely COVID driven, or where there also pregnancy related indications ? where all cases included ?
Answer: Thank you for the well-stated comment. A new paragraph was added in the methods section. The clinical symptoms for admission were in accordance with the local COVID-19 protocol, including need for respiratory support, fever above 38oC for more than 24 hours, confirmed pneumonia etc. Asymptomatic for COVID-19 infected pregnant women that were hospitalized for obstetrical indications were excluded from the present study in order to avoid systematic bias.
Reviewer comment: Was the scoring system used prospectively, or retrospectively, and the assessors were likely not blinded for covid, hospitalization and outcome ? and how were settings handled with several ultrasounds (the worst imaging, the highest score, progressing pregnancy). How should we compare the findings in hospitalized and non-hospitalized cases (80 and 25 %), to ‘reference datasets over gestational age’ ?
Answer: The scoring system has been used retrospectively by the same examiner (Department of Prenatal Ultrasound). We have been used the same standard settings in all cases. All cases (hospitalized or not) were followed-up in our unit as a part of our standard obstetrical care.
Reviewer comment: Are there any data on pregnancy outcome, and on APO of the placenta after delivery ?
Answer: We would like to thank the reviewer for the well-stated comments. The present cohort study includes recently collected date between September to December 2021 and the most of the pregnancies included have not been terminated yet. Some of the placentas were examined macrospopically and confirmed the ultrasound findings.
Reviewer comment: Editing:
I think that the reference numbers should be adapted to be in line with guidelines of the journal.
Answer: Thank you for the comment. Reference numbers were modified accordingly.
Round 2
Reviewer 1 Report
Dear authors,
You must write that the scoring system was created by your team in order to be simple and reproducible for further research.
Best regards
Author Response
We would like to thank you for your well-stated comments.
As you suggested, in methods sections we report "A scoring system of all placental ultrasonographic findings that were observed was created by our team in order to be simple and reproducible for further research".
Reviewer 2 Report
there are still limitations on this report (retrospective, no outcome data, no valid scoring system), but these limitations are acknowledged.
Author Response
We would like to thank the reviwer for the well-stated comments. We have updated the limitation paragraph in accordance to your comments.